# An Improved Isotope Labelling Method for Quantifying Deamidated Cobratide Using High-Resolution Quadrupole-Orbitrap Mass Spectrometry

**DOI:** 10.3390/molecules27196154

**Published:** 2022-09-20

**Authors:** Bo Liu, Lu Huang, Rongrong Xu, Huihong Fan, Yue Wang

**Affiliations:** 1National Institutes for Food and Drug Control, 31st Huatuo Rd., Daxing Dist., Beijing 102629, China; 2NMPA Key Laboratory for Quality Research and Evaluation of Chemical Drugs, Beijing 102629, China

**Keywords:** cobratide, deamidation impurity, high-resolution mass spectrometry, ^18^O-labelling

## Abstract

Protein deamidation can severely alter the physicochemical characteristics and biological functions of protein therapeutics. Cobratide is a non-addictive analgesic with wide clinical acceptance. However, the Asn residue at position 48 from the N-terminus of the cobratide amino acid sequence (N48) tends to degrade during purification, storage, and transport. This characteristic could severely affect the drug safety and clinical efficacy of cobratide. Traditional methods for quantitating deamidation reported in previous research are characterised by low efficiency and accuracy; the quality control of cobratide via this method is limited. Herein, we developed an improved ^18^O-labelling method based on the detection of a unique peptide (i.e., the protein fragment of cobratide containing the N48 deamidation hotspot after enzymolysis) using an Orbitrap high-resolution mass spectrometer to quantify deamidated cobratide. The limits of detection and quantification of this method reached 0.02 and 0.025 μM, respectively, and inter- and intra-day precision values of the method were <3%. The accuracy of the ^18^O-labelling strategy was validated by using samples containing synthesised peptides with a known ratio of deamidation impurities and also by comparing the final total deamidation results with our previously developed capillary electrophoresis method. The recoveries for deamidation (Asp), deamidation isomerisation (iso-Asp), and total deamidation were 101.52 ± 1.17, 102.42 ± 1.82, and 103.55 ± 1.07, respectively. The robustness of the method was confirmed by verifying the chromatographic parameters. Our results demonstrate the applicability of the ^18^O-labelling strategy for detecting protein deamidation and lay a robust foundation for protein therapeutics studies and drug quality consistency evaluations.

## 1. Introduction

Post-translational modifications (PTMs), such as deamidation (e.g., Asn to Asp) and deamidation–isomerisation (e.g., Asn to isoAsp), have tremendous effects on the high-level structure, kinetic stability, and biological function of functional proteins and protein therapeutics [1,2,3,4,5,6,7]. Furthermore, PTM is the most common degradation pathway for protein therapeutics. During this process, the side-chain carbonyls of Asn residues in a protein sequence are converted into a succinimide intermediate and then hydrolysed into Asp and isoAsp [8].

The rate of deamidation–isomerisation has been demonstrated to be especially high when the C-terminus of the Asn residue is connected to a Gly residue [9]. Thus, protein therapeutics of this type tend to generate deamidation impurities during production and storage.

Cobratide, a protein component isolated from the venom of Chinese cobra *(Naja atra)* [10], is exceptionally effective in relieving chronic, long-term, and refractory pains, especially tumour-related pains. Owing to its extremely low addictiveness, cobratide has a wide range of clinical applications [11], particularly as an analgesic for children. The protein was also recently discovered to exhibit therapeutic potential for COVID-19 [12]. The structure of cobratide includes a deamidation hotspot at position 48 from the N-terminus (N48), where the C-terminus of an Asn residue is connected to a Gly residue. Thus, developing a highly accurate method to quantify deamidation impurities in cobratide is of great importance in ensuring consistency in its manufacturing and storage.

At present, high-resolution mass spectrometry (MS) remains the most powerful tool for determining heterogeneity in protein samples; as such, tremendous efforts have been devoted to expanding the applications of MS in qualitative and quantitative research [13,14,15,16]. However, the direct determination of the extent of protein deamidation using the top-down strategy is challenging and cannot be achieved by MS or chromatography. In MS, the mass difference between a protein and its deamidation mutant is only 0.98 Da, and the protein is usually multi-charged. Moreover, the peak with the highest signal in a cobratide sample has a charge of 5 or 6, which means the nearest isotope peak between cobratide and its deamidated variant is only approximately 0.003 Da. In chromatography, only one amino acid differs between cobratide and its deamidated variant. Thus, obtaining good resolution between the protein and its two deamidation variants (i.e., deamidated (Asn→Asp) and isomerised–deamidated (Asn→isoAsp)) is difficult. In previous work, we successfully separated cobratide and its deamidation components via capillary electrophoresis (CE)–UV spectroscopy; however, we could not achieve the separation of the deamidated and isomerised–deamidated proteins [17]. These issues explain why the precise rates of deamidation (Asn→Asp) and isomerisation–deamidation (Asn→isoAsp) of the protein are usually quantified using a bottom-up approach. In this approach, the protein is first denatured and then digested to facilitate the analysis of deamidation. However, steps such as denaturation and enzymolysis can cause artificial deamidation, which induces false positives. Several analytical methods based on the optimisation of digestion conditions have been established to solve this problem [18,19,20,21,22]. In a previous study, we successfully improved the accuracy of deamidation impurity analysis by modifying the digestion conditions [22]. However, this method can only limit the introduction of deamidation impurities during enzymatic digestion; it cannot prevent deamidation due to other artificial factors prior to enzymatic digestion (e.g., deamidation due to protein denaturation or reductive alkylation). Therefore, quantifying the true extent of deamidation in a protein sample is a worthwhile endeavour.

Liu et al. developed an analytical approach based on ^18^O labelling using H_2_^18^O as the solvent throughout the sample preparation process [23,24]. Time-of-flight (TOF) MS was then employed to differentiate the inherent deamidation of the sample from its artificial deamidation. This method established a foundation for the precise determination of deamidation ratios in protein samples. However, the use of trypsin for enzymatic digestion generates a large number of cleavage sites, which often results in multiple protein fragments with the same deamidation site due to missing and erroneous cuts. This issue increases the difficulty of calculating the exact extent of deamidation in a protein sample. In addition, the lack of a validation strategy for this method limits its application scope, especially in new drug applications for biopharmaceuticals and quality consistency evaluations of biosimilar and generic drugs. These problems can be resolved as follows: (1) the analytical conditions of the method, including the sample pre-treatment, data acquisition, and data analysis procedures, must be improved; (2) the necessary calculations must be simplified by improving the sample preparation process to obtain a unique peptide with the deamidation site (i.e., a single peptide containing the deamidation site after enzymolysis) so that the rate of deamidation of cobratide could be calculated using only this unique peptide and the ^18^O-labelled unique peptide (i.e., the unique peptide incorporated with ^18^O at the –OH group of the C-terminal of this fragment) after enzymolysis in H_2_O and H_2_^18^O, respectively; (3) accuracy must be enhanced by improving the resolution of isotopic peaks so that the false positive peak could be eliminated; and (4) a rational and thorough method validation strategy must be formulated. Considering these points, we developed an improved method that employs quadrupole-Orbitrap tandem high-resolution MS (THRMS) to quantify deamidation impurities in cobratide samples based on the ^18^O labelling of a unique peptide. Our method could improve the accuracy of protein deamidation analysis and expand the applicability of the ^18^O-labelling strategy in terms of instrumentation and range of applications. The findings in this study could serve as a reliable reference for subsequent studies on quality consistency for other protein therapeutics.

## 2. Results

### 2.1. Glu-C for Enzymatic Digestion

In the bottom-up strategy, the availability of several peptides containing some PTM site would theoretically require the inclusion of the PTMs of all of these peptides when calculating the relative abundance of PTMs in the protein sample. Cobratide contains many K and R residues. When we used trypsin for enzymatic cleavage in a previous study, we did not obtain a unique peptide for the N48 site (Figure 1a). Therefore, in this study, we used Glu-C protease for enzymatic cleavage. This modification allowed us to obtain the unique peptide fragment for positions 39–51 (RGCGCPSVKNGIE) containing the N48 site of the cobratide amino acid sequence (Figure 1b).

### 2.2. Introduction to the Calculation

Orbitrap THRMS was used to analyse samples prepared in H_2_O and H_2_^18^O. Biopharma Finder software was used to determine the retention time and charge number of the R39-E51 peptide; note that the retention time of the isotope-labelled peptide is essentially identical to that of the unlabelled peptide. The triply charged peaks of this peptide in the normal water samples produced the strongest responses. The monoisotopic masses obtained were Asn-^16^O: *m/z* = 478.5576 and isoAsp-^16^O and Asp-^16^O: *m/z* = 478.8857). By contrast, the corresponding triply charged peptides in the H_2_^18^O-treated samples had monoisotopic masses of Asn-^18^O: *m/z* = 479.2256 and isoAsp-^18^O and Asp-^18^O: *m/z* = 479.5541. The mass spectra of these ions are shown in Figure 2.

### 2.3. Calculation of the Percentages of Asp and isoAsp Contents

The steps below describe how the percentages of Asp and isoAsp contents are calculated.

#### 2.3.1. Calculation of Total Deamidation

The first step in the proposed THRMS-based ^18^O-labelling method is the quantification of the total deamidation impurities of the sample (Asn → Asp and Asn → isoAsp), including both inherent and preparation-induced deamidation impurities. As is shown in Figure 3, We summed the extracted ion chromatograms (EICs) for all isotopic peaks of the three peptides to obtain the total signal strength of each peptide and improve the selectivity of Asn, isoAsp, and Asp (the peaks used are shown in Appendix A). By calculating the ratios between the total peak areas of the isotopic peaks of these peptides, we can obtain the total deamidation ratios of Asp and isoAsp, including the intrinsic deamidation of the sample and the artificial deamidation introduced by the sample preparation process.

#### 2.3.2. Calculation of Isotope Overlaps

In the sample prepared in normal water (i.e., the ‘H_2_O sample’), the monoisotopic peak (peak 1) is more intense than peak 3. In the sample prepared in ^18^O-enriched water (i.e., the ‘H_2_^18^O sample’), peak 3 is more intense than peak 1 (Figure 2b, at *m/z* = 479.8929 and *m/z* = 479.2256). According to Liu et al. [23,24], as illustrated in Figure 4, the signal of peak 3 in the H_2_^18^O samples consists of: (1) the natural ^18^O abundance of the peptide, (2) the signal arising from the incorporation of one ^18^O at the C-terminus, and (3) the ^18^O introduced to the sample by deamidation during sample preparation. Therefore, the overlaps by (1) and (2) must be solved to calculate the percentage of ^18^O introduced during sample preparation.

##### 2.3.2.1. Determination of the Overlap by the Natural Isotopic Abundance

As shown in Figure 2a (Asn-^16^O), Peak 1 in the mass spectrum represents the monoisotopic peak of the non-deamidated R39-E51 fragment in normal H_2_O, while peak 3 represents the isotopic distribution of the substance with the monoisotopic signal of peak 1. In other spectra of Figure 2, peak 3 is overlapped by the same ratio of natural isotopic abundance in Figure 2a (Asn-^16^O). Therefore, the ratio of overlap by the natural isotopic abundance in the deamidated peak could be calculated by the ratio of peak 3 to peak 1 in the mass spectrum of Asn-^16^O in normal H_2_O (see in Figure 2a, Asn-^16^O).

##### 2.3.2.2. Determination of the Overlap by Peptides with Two C-Terminal ^18^O Atoms

Because the peptide samples were prepared in ^18^O-enriched water, ^18^O is incorporated at the –OH group of the C-terminus, which increases the molecular weight of the peptides by 2 (m + 2, with a difference of 0.66 in triply charged species). In Section 2.3.2.1, we calculated the deamidation ratio of the sample in normal water. Therefore, the overlap due to ^18^O incorporation at the C-terminus can be calculated by dividing peak 3 (*m/z* = 479.2259) by peak 1 (*m/z* = 479.8929) in the Asn-^18^O fragment (Figure 2b) and then subtracting the overlap calculated in Section 2.3.2.1.

Note that the integrated peak areas of the extracted isotopic ion currents (Figure 3c,e) were used in these calculations to maximise precision.

#### 2.3.3. Calculation of the Deamidation Ratio Prior to Sample Preparation

In the H_2_^18^O sample, the total Asp signal of the peptide incorporating one C-terminal ^18^O atom is given by the sum of peak 1 (*m/z* = 479.554) and peak 3 (*m/z* = 480.221) in the Asp-^18^O and isoAsp-^18^O peptides (Figure 4) minus the two overlaps calculated in Section 2.3.2. For the Asp-^18^O and isoAsp-^18^O peptides, peak 1 represents the deamidation of Asn prior to sample preparation. Therefore, the percentage of total deamidation due to the deamidation of the cobratide sample prior to sample preparation is given by the area of peak 1 divided by the total deamidation (peak 1 + peak 3 − sum of overlaps from Section 2.3.2) × 100.

#### 2.3.4. Correction of the Deamidation Ratio

Because obtaining 100% pure H_2_^18^O water is not possible, the calculations must be corrected to account for the presence of normal water. The isotopic purity of the H_2_^18^O water was 98.0%, and the corrected result was calculated as (result from Section 2.3.3 − (100% − result from Section 2.3.3) × 0.02/0.98).

#### 2.3.5. Calculation of the Relative Percentage of Asp/isoAsp Present in the Sample Prior to Sample Preparation

The relative percentage of Asp/isoAsp could be obtained by multiplying the peak areas of Asp, isoAsp, and total deamidation in Section 2.3.1 with the percentage obtained in Section 2.3.4. The total deamidation obtained was compared with that determined by the CE–UV method we had previously developed and the deviation between the two methods was less than 1% (Table 1).

### 2.4. Validation

The proposed method was validated by calculating its linearity, precision, accuracy, and limit of quantification (LOQ) according to the International Conference on Harmonization (ICH) guidelines [25].

#### 2.4.1. Specificity

The specificity of the proposed method refers to its ability to accurately measure the extent of deamidation in the presence of a blank solution. The results indicated that the matrix solution did not interfere with quantification of deamidation.

#### 2.4.2. Precision

The inter- and intra-day precision values of the proposed method over two consecutive days were evaluated by calculating the relative standard deviation (RSD) of six sets of sample solutions prepared from sample S1, as shown in Table 2. The inter- and intra-day precision values of the method were both lower than 3%.

#### 2.4.3. Limit of Detection (LOD) and Limit of Quantification (LOQ)

The sensitivity of the proposed method depends on whether the signal of the R39-E51 peptide could be quantified or detected after enzymolysis. Because the test sample contains different levels of deamidation, LOD and LOQ were described using the data of three synthesised peptide fragments, namely, T37-N53 (Asn), T37-N53 (Asp), and T37-N53 (isoAsp). After the enzymolysis of a series concentration of mixed solutions ranging from 0.015 to 0.15 µM. the lowest signal-to-noise ratio (S/N) of all signals of the three peptide fragments (Asn, Asp, and isoAsp) could be used to determine the LOD and LOQ, which were found to be 0.02 µM (S/N = 5) and 0.025 µM (S/N = 16), respectively.

#### 2.4.4. Robustness

The area ratio of Asp and isoAsp may vary when the liquid chromatographic parameters fluctuate. Thus, the effect of several parameters, such as flow rate, injection volume, column temperature, and mobile phase ratio, was studied. Differences in these parameters did not lead to marked variations in the area ratios (Table 3).

#### 2.4.5. Recovery

Because preparing cobratide and its deamidation degradant as single components is difficult, we used three synthesised peptide fragments containing the N48 site to evaluate the accuracy of our method. The percentage recoveries of IsoAsp, Asp, and total deamidation were found to be 101.52%, 102.42%, and 103.55% (Table 4), respectively. These values confirm the accuracy of the developed method. The use of peptide fragments with content data is necessary to measure recovery (Figure 5). To this end, we used the T37-E53 amino acid sequence of cobratide (i.e., T37-N53_Asn_, T37-N53_Asp_, and T37-N53_isoAsp_), as shown in Figure 5. These fragments were subjected to enzymatic digestion in the sample solution, followed by MS analysis. The recoveries of these characteristic peptides are shown in Table 3.

## 3. Discussion

In bottom-up proteomics, the relative abundance of PTMs can only be calculated by summing the PTMs of all peptide fragments containing the mutational site. As the isotopic labelling method has yet to be automated, the calculation of relative PTM abundances using this approach is highly time- and effort-intensive. In this work, we obtained unique peptides for the deamidation hotspot of cobratide by optimising the enzymatic digestion process. This step allowed for the quick and accurate calculation of the deamidation ratio of cobratide.

Because a method to validate deamidation ratios using the ^18^O-labelling strategy has yet to be reported, verifying the accuracy of this method is not yet possible. In this work, three synthetic polypeptide fragments containing the deamidation hotspot of cobratide (Asn: T37-N53_asn_; Asp: T37-N53_asp_, isoAsp: T37-N53_isoAsp_) were introduced to the recovery calculation process at predetermined proportions to enable the calculation of the deamidation ratio of cobratide. This step allowed us to validate the proposed method.

Because the chromatographic behaviours of deamidated and non-deamidated peptides do not differ significantly, difficulties in completely resolving Asn and Asp in some peptide fragments, such as R39-E51, were encountered in this study (Figure 3a). Therefore, the ion currents of the isotopic signals of Asn, Asp, and isoAsp were extracted, and the peak areas of these signals were summed. This step enabled us to calculate the total deamidation ratio accurately even when the peptide peaks were not fully separated by TIC. Furthermore, the ultra-high resolution of the Orbitrap THRMS instrument for isotopic peaks was used to calculate the ion currents of the Asn, Asp, and isoAsp polypeptides, which are critical for calculating the total deamidation of cobratide before and after sample preparation. This modification means the proposed method could be performed using a larger variety of instruments, which is certainly advantageous as previous isotopic labelling methods could only be performed using TOF mass spectrometers, for a wider range of applications. Furthermore, the optimisation of chromatographic conditions, which is well known to be a tedious process, is no longer necessary in our method, which greatly increases its development efficiency.

## 4. Materials and Methods

### 4.1. Chemicals and Reagents

The cobratide used in this study was produced by Yun Nan Nanzhao Pharmaceutical Co., Ltd. (Dali, China). Ammonium bicarbonate, iodoacetamide, urea, and dithiothreitol (DTT) were purchased from Sigma-Aldrich (St. Louis, MO, USA). 18O-enriched water (98.0% purity) was obtained from CIL (Cambridge, UK). MS-grade trypsin (20 µg), acetonitrile, formic acid, and trifluoroacetic acid (TFA) were purchased from Thermo Fisher (Waltham, MA, USA).

### 4.2. Preparation of the Test Solution

The cobratide samples were first dissolved in 100 mM phosphate buffer with ^18^O-enriched water to achieve a concentration of 10 mg/mL. The sample was then denatured, reduced at 56 °C for 30 min using 10 mM DTT in the presence of 1.2 M guanidine chloride in ammonium bicarbonate buffer, and treated with 20 mM iodoacetamide at room temperature for 30 min in the dark to alkylate the free thiol groups. The buffer in the sample was then exchanged with 100 mM phosphate buffer prepared in ^18^O-enriched water using an ultrafiltration centrifuge tube. Glu-C was added to each sample solution at a concentration ratio of 1:10 (Glu-C/protein) and then incubated overnight at 37 °C. A normal water control was prepared by treating the samples as described above using buffers dissolved in ^16^O water.

### 4.3. Liquid Chromatography/Mass Spectrometry

An Ultimate 3000 UHPLC device (Thermo Fisher, Waltham, MA, USA) equipped with an XBridge BEH C18 column (2.1 × 150 mm, 2.5 μm particle size, Waters, Milford, MA, USA) was coupled with a Q-Exactive Orbitrap high-resolution mass spectrometer (Thermo Fisher) to analyse the peptides. Approximately 10 μg of each sample was injected into the column at an initial condition of 98% mobile phase A (0.1% formic acid in water) and 2% mobile phase B (0.1% formic acid in acetonitrile). The peptides were eluted by first increasing the concentration of mobile phase B to 32% within 42 min and then to 90% mobile phase B within 5 min. The column was washed using 100% mobile phase B and then equilibrated using 2% mobile phase B prior to the next injection. The column oven was set to 45 °C, and the flow rate was 0.2 mL/min. The mass spectrometer was operated in full-scan mode from *m/z* 200 to 1500 coupled with full MS/dd-MS^2^ in positive-ion mode.

### 4.4. Validation Parameters of the Proposed Method

The proposed method was validated according to the ICH guidelines by calculating its specificity, precision (repeatability and intermediate precision), accuracy (recovery), LOD, and LOQ.

#### 4.4.1. Specificity

Blank buffer was prepared and analysed for validation of the method specificity.

#### 4.4.2. Precision

Precision was determined by calculating the repeatability (intra-day precision) and intermediate precision (inter-day precision) of the method. S1 samples were prepared according to the procedure described above (*n* = 6) and analysed according to the proposed method (for intra-day precision) over two consecutive days (for inter-day precision).

#### 4.4.3. Accuracy

Accuracy was expressed as the percentage recovery for each parameter, including Asp, isoAsp, and the total deamidation of cobratide in the S1 sample with three peptides (T37-N53_Asn_, T37-N53_Asp_, and T37-N53_isoAsp_). A mixed recovery solution (T37-N53_Asn_:T37-N53_Asp_:T37-N53_isoAsp_ = 90:7:3) was prepared to evaluate accuracy.

#### 4.4.4. Sensitivity

The LOD and LOQ were determined based on the signal-to-noise ratio (S/N). In this approach, a series of test solutions with different concentrations was prepared, and the LOD and LOQ were determined at S/N ratios of 3:1 and 10:1, respectively.

#### 4.4.5. Robustness

The liquid chromatographic parameters (i.e., flow rate, injection volume, column temperature, and mobile phase ratio) were varied, and their effect was evaluated in terms of the peak area ratio. If the RSD (%) was <2%, the method was considered to be robust.

## 5. Conclusions

A novel method was developed for the precise quantification of the deamidation ratio of cobratide. This method uses an Orbitrap THRMS instrument with an improved ^18^O-labelling strategy based on a unique peptide. A validation method in which synthetic characteristic peptides of a specific deamidation site were added at predetermined ratios was also designed for the proposed strategy. This method greatly increased the credibility of the proposed analytical strategy and paves the way for this strategy to become the standard quality control method for protein therapeutics. Studies on the use of the proposed quantification method for other protein therapeutics are currently under way. The development of computational techniques and automation software for this method is also being explored.

## Figures and Tables

**Figure 1 molecules-27-06154-f001:**
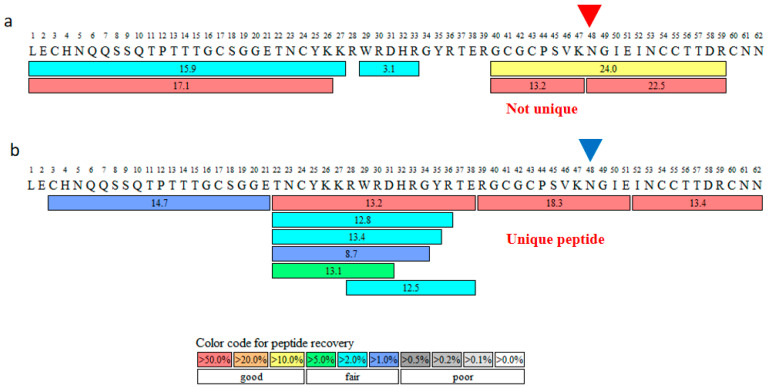
Enzymatic cleavage by (**a**) trypsin and (**b**) Glu-C protease. Enzymatic cleavage with Glu-C protease gives a unique peptide that includes the N48 site. Peptide recovery was calculated using Biopharma Finder software, which defines the concept as the percentage of the total peak area for a given peptide (modified and unmodified forms) relative to the self-weighted average of all components. Thus, a recovery of >10% is considered good recovery.

**Figure 2 molecules-27-06154-f002:**
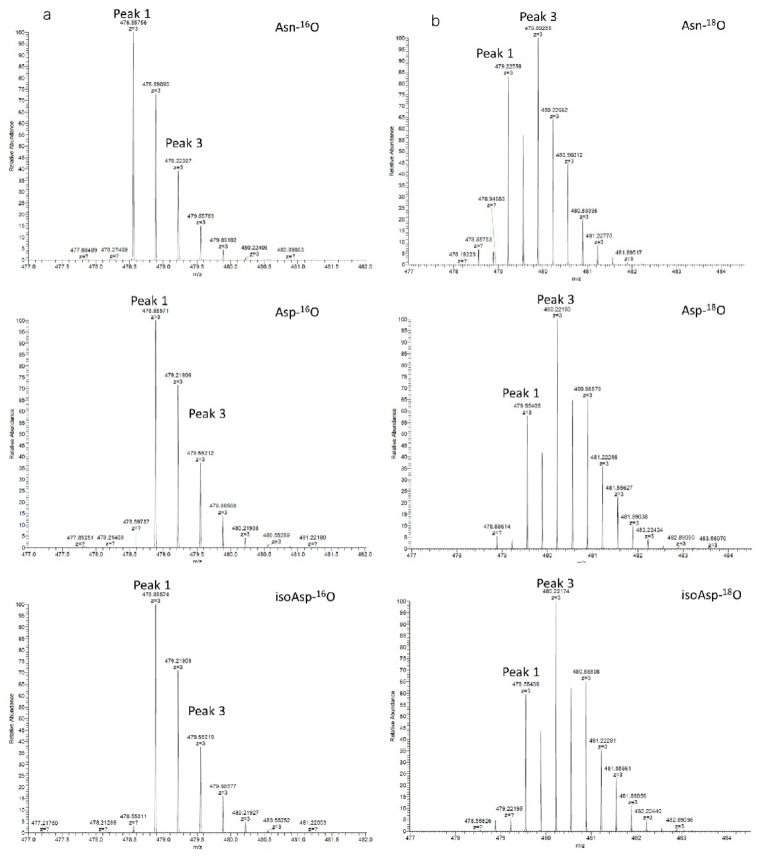
Mass spectra of the triply charged peptide R39-E51 obtained from digestion in normal water (^16^O, **a**) and ^18^O-enriched water (^18^O, **b**). The mass spectra are labelled accordingly.

**Figure 3 molecules-27-06154-f003:**
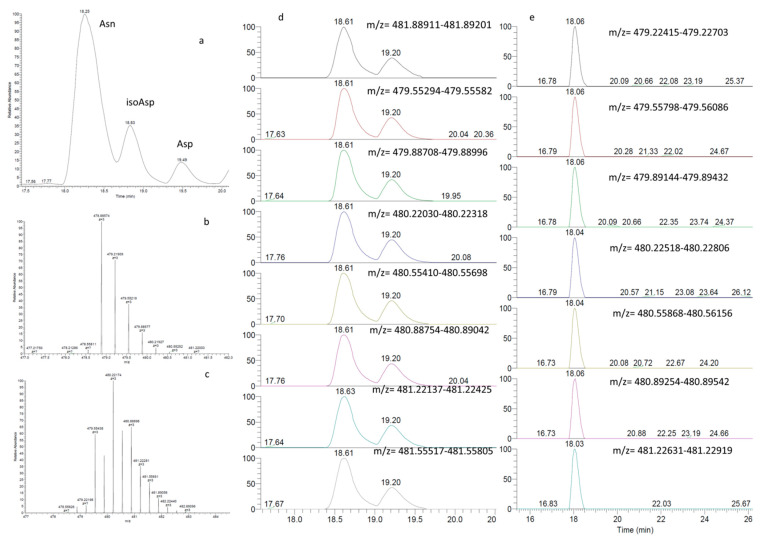
TIC and EICs of Asn, isoAsp, and Asp. (**a**) TIC plots of Asn, isoAsp, and Asp. (**b**) Mass spectrum of the Asn-^18^O peak in the isotopically labelled sample. (**c**) EIC of the isotopic peaks from Asn-^18^O in the ^18^O-labelled sample. (**d**) Mass spectrum of Asp-^18^O and isoAsp-^18^O in the isotopically labelled sample. (**e**) EIC of the isotopic peaks from Asp and isoAsp in the ^18^O labelled sample.

**Figure 4 molecules-27-06154-f004:**
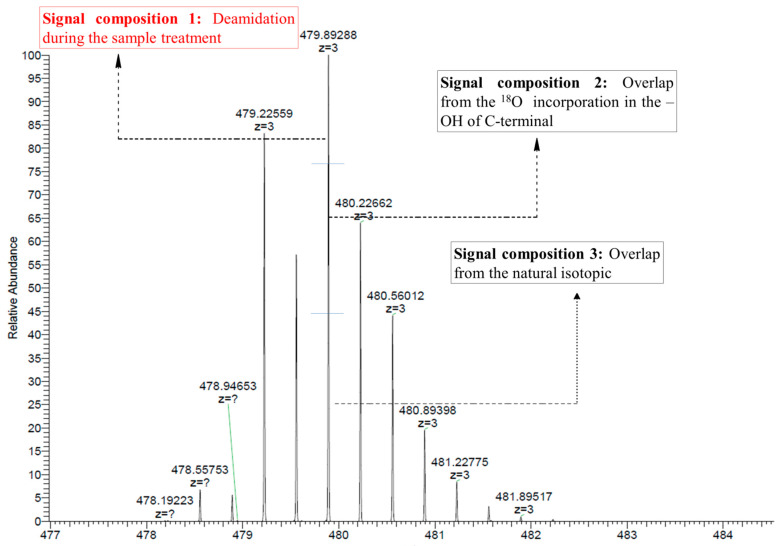
Signal components of peak 3 from the Asp-^18^O peptide. Peak 3 contains three components: the signal of deamidation during the sample treatment, the overlap from the ^18^O incorporated into the –OH group of the C-terminal (obtained in Section 2.3.2.2), and the overlap from the natural isotopic abundance (obtained in Section 2.3.2.1).

**Figure 5 molecules-27-06154-f005:**
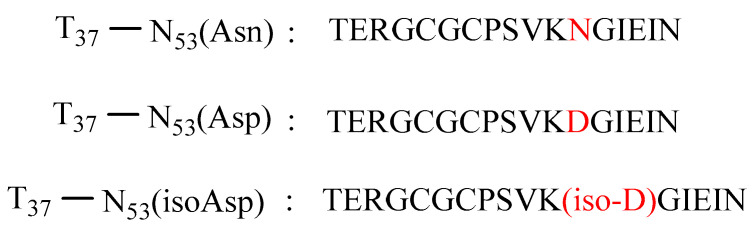
Sequence of three synthetic peptides used to measure recovery.

**Table 1 molecules-27-06154-t001:** Comparison of the quantification results of isoAsp and Asp determined in five batches of samples using both the proposed method and our previously reported CE–UV method.

Sample	Deamidation Result (Proposed Method)			Deamidation Result (CE–UV) ^a^
	isoAsp	Asp	Total Deamidation	Total Deamidation
S1	12.0%	4.3%	16.3%	15.8%
S2	17.8%	9.0%	26.8%	25.9%
S3	16.9%	7.4%	24.3%	24.9%
S4	4.7%	2.3%	7.0%	8.1%
S5	13.1%	6.0%	19.1%	18.3%

^a^ Total deamidation determined by the CE–UV method we had previously reported [17].

**Table 2 molecules-27-06154-t002:** Intra- and inter-day precision of the improved isotope labelling method for sample S1.

	RSD (%)	
Impurity	Intra-Day ^a^	Inter-Day ^b^
IsoAsp	1.6%	2.8%
Asp	2.0%	2.2%
Total Deamidation	1.7%	2.1%

^a^ *n* = 6 for each sample; ^b^
*n* = 12 for each sample over two consecutive days.

**Table 3 molecules-27-06154-t003:** The robustness of the method.

NO	Parameter	Level ^a^	Area Ratio	
			Mean (%) ^a^	RSD (%)
1	Flow rate	A	1.54	0.0
		B	1.54	0.0
2	Mobile phase	A	1.57	1.3
		B	1.55	0.4
3	Column oven temperature	A	1.56	0.0
		B	1.55	0.9
4	Injection volume	A	1.55	0.9
		B	1.54	0.0

^a^ The upper (A) and lower (B) levels of the robustness parameters are shown as follows: flow rate: A = 0.18 mL/min, B = 0.22 mL/min; mobile phase: A (content of formic acid) = 0.09%, B (content of formic acid) = 0.11%; column oven temperature: A = 43 °C, B = 47 °C; injection volume: A = 9 μL, B = 11 μL.

**Table 4 molecules-27-06154-t004:** Calculated recoveries ^a^.

Impurity	Recovery (%)	RSD (%)
IsoAsp	101.52 ± 1.17	1.15
Asp	102.42 ± 1.82	1.78
Total Deamidation	103.55 ± 1.07	1.03

^a^ *n* = 6 for each impurity.

## Data Availability

The data presented in this study are available on request from the corresponding author.

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
