# Peer review of "An Improved Isotope Labelling Method for Quantifying Deamidated Cobratide Using High-Resolution Quadrupole-Orbitrap Mass Spectrometry"

_molecules, 2022, doi:10.3390/molecules27196154_

Round 1

Reviewer 1 Report (Previous Reviewer 1)

The authors addressed most of the comments and the revised version can be accepted.

Author Response

Thank you for your supporting.

Reviewer 2 Report (Previous Reviewer 3)

The authors made important improvements to the manuscript. However, several additional changes can improve the manuscript even further.

Title

1)      “precise quantification” has almost no meaning in analytical chemistry if it is not accurate.

Abstract

2)      It is unlikely that one of the following values is correct “The limits of detection and quantification of this method could reach 0.02 and 0.025 μM,”. In typical measurements LODs and LOQs have higher differences. Please recheck.

3)      The same as in the title “precisely quantify deamidated”. Add values to demonstrate accuracy of measurements.

4)      Redundancy of statements in case of “The proposed method achieved the precise 24 quantification of deamidation impurities in cobratide.” And “using an Orbitrap high-resolution mass spectrometer was developed to precisely quantify deamidated cobratide.” Can be shortened.

5)      No information on robustness and repeatability of developed method presented.

Results

6)      What is the recovery in Fig. 1 legend? Clarifications are needed.

7)      Not sure what the following means “2.2. Introduction of the peptide to the calculations”.

8)      More specificity is needed for following “We summed the extracted ion chromatograms (EICs) for all isotopic peaks of the three peptides”. There are a number of overlapping monoisotopic patterns especially complex in cases of Asp-18O and iso-Asp-18O (Fig. 2). Which peaks were summed? Used peaks and their m/zs as well assignments should be listed in separate table in supplementals.

9)      Define “natural isotopic peak” in “Peak 1 in the mass spectrum of the non-deamidated R39-E51 fragment in H2O is the  monoisotopic peak, while peak 3 is the natural isotopic peak.” Peak 3 is a sum of two monoisotopic signals - 3rd peak in isotopic distribution of substance with monoisotopic signal named peak 1 and monoisotopic peak of substance named peak 3. Presented description in 2.3.2.1. is somewhat useful but requires rewriting. Following should be well illustrated “Therefore, the ratio of overlap of the deamidated peak with the natural isotopic  abundance may be calculated by dividing peak 3 with peak 1 in the mass spectrum of the  non-deamidated R39-E51 fragment in H2O (Asn-16O).”. Does Fig. 4 relates to this description?

10)   Figure 4 legend should be expanded. Currently, it is difficult to understand what is shown.  It looks like that one signal represents three different molecules.

11)   Data on the determination LODs and LOQs have to be presented. How linear were calibration curves and how well experimental points were aligned? 

Round 2

Reviewer 2 Report (Previous Reviewer 3)

The authors made important adjustments to the manuscript making the presentation of results sufficiently clear for publication.

This manuscript is a resubmission of an earlier submission. The following is a list of the peer review reports and author responses from that submission.

Round 1

Reviewer 1 Report

The authors reported a method for quantification of deamidation impurities in cobratide using 18O-labelling mass spectrometry. The submission is unacceptable for the following points:-

1.      The novelty of the study is low. The motivation of 18O-labelling is not well-addressed.

2.      Experimental part including Materials and methods, sample preparation, …etc are missed.

3.      Orbitrap high-resolution mass spectrometer is enough for direct qualitative and quantification analysis of biomolecules. The authors have to motivate their strategy.

4.      The title should be revised to be clear, precise and informative. Information such as ‘mass spectrometry’ should be highlighted.

5.      A comparison with previously published methods should be discussed and summarized into a Table.

6.      Introduction should be rewritten. A brief introduction of mass spectrometry should be included.

7.      References for MS should be updated including these References; https://doi.org/10.1016/j.trac.2014.09.010; https://doi.org/10.1038/s41592-021-01197-1; https://doi.org/10.1021/acs.chemrev.1c00696

8.      The language should be revised and typos should be corrected.

  Minors

9.      ‘et al.’ should be italic.

Reviewer 2 Report

In the manuscript “Precise quantification of deamidation impurities in cobratide using an improved strategy based on 18O-labelling and unique peptides”, the authors describe the use of Glu-C protease for enzymatic cleavage to generate unique peptide fragment for the positions containing the N48 site of the cobratide amino acid sequence.  An LC-HRMS method, associated to 18O-labelling strategy, was used to quantify deamidation impurities in cobratide.

-          Section 2.2: Selection of enzymatic digestion conditions: the text does not match the section title.

-          Lines 115-117 (section 2.3): “because of retention times of Asn, isoAsp, and Asp are similar in the TIC, completely separating their peaks is rather challenging” - some modifications in the gradient used in the chromatographic method could have been evaluated to improve the resolution of the method. From my point of view, this sentence doesn't make sense: retention times are similar because you haven't optimized the separation conditions. If it is not the focus of the work, just say that you used the EIC for all isotopic peaks of the three peptides for better selectivity.

-          Lines 133-135 – insert “as illustrated in Figure 4”.

-          Figure 4: “deamidation” is underlined

-          Lines 176-177: Which guideline did you consider in the validation step?

-          Insert more details regarding the method validation in the experimental section.

-          There are some misspellings in the “author contributions” 

  Try to make a final revision of the text, trying to make the sentences clearer

Reviewer 3 Report

Manuscript “Precise quantification of deamidation impurities in cobratide using an improved strategy based on 18O-labelling and unique peptides” presented by Bo Liu describes an approach for determination of ratios of amidated and deamidated cobratide using LC-MS. Described information represent an example of application of previous published approach ( https://www.sciencedirect.com/science/article/pii/S000326971200471X?via%3Dihub ) to a new analyte.   Despite of adaptation of this approach to a different analysis system involving high mass resolution instrument, this work does not possess enough novelty to be published as a separate article.

 Several comments which may improve this manuscript after its significant enhancement with new information are below.

Title

1)    Title can be improved. For example, “deamidation impurities of cobratide” correspond to deamidated cobratide. This is simpler and more clear expression. “unique peptides” are also difficult to link to the topic. Are these peptides fragments of deamidated cobratide?

Abstract

2)    Abstract has similar challenges as the title. There is no info unique peptides or important protein residues labeled with oxygen isotope. No information of efficiency of labeling and LOQ of described approach. Addition of such information will improve abstract.

Introduction

3)    14 references for intro statement for research article is too much unless all these references are used in following parts of the manuscript - “[1–14]”.

4)    Introduction also does not clarify what unique peptides and how they are labeled e.g. before or after their formation.

5)    For some reason, another publication on cobratide impurities quantitative analysis published by the same group is not referenced and discussed https://pubs.rsc.org/en/content/articlelanding/2021/ay/d1ay00717c

Results

6)    Fig. 1. Peptide recovery should be explained in legend. Also, statistical information should be added for peptide recovery of key peptides. >50% is too vague.